# Long-Term High-Fat Diet Limits the Protective Effect of Spontaneous Physical Activity on Mammary Carcinogenesis

**DOI:** 10.3390/ijms25116221

**Published:** 2024-06-05

**Authors:** Sébastien Marlin, Marie Goepp, Adrien Desiderio, Stéphanie Rougé, Sahar Aldekwer, Delphine Le Guennec, Nicolas Goncalves-Mendes, Jérémie Talvas, Marie-Chantal Farges, Adrien Rossary

**Affiliations:** 1UNH—Unité de Nutrition Humaine, CRNH-Auvergne, Université Clermont-Auvergne, INRAe, F-63000 Clermont-Ferrand, France; sebastien.marlin@uca.fr (S.M.); adrien.desiderio@laposte.net (A.D.); stephanie.rouge@uca.fr (S.R.); sahardekwer93@gmail.com (S.A.); delphine.r.le.guennec@gmail.com (D.L.G.); nicolas.goncalves-mendes@uca.fr (N.G.-M.); jeremie.talvas@uca.fr (J.T.); 2Resolution Therapeutics, University of Edinburgh, Edinburgh EH16 4UU, UK

**Keywords:** high-fat diet, obesity, spontaneous physical activity, mammary carcinogenesis, tumour microenvironment, immunity, oxidative stress

## Abstract

Breast cancer is influenced by factors such as diet, a sedentary lifestyle, obesity, and postmenopausal status, which are all linked to prolonged hormonal and inflammatory exposure. Physical activity offers protection against breast cancer by modulating hormones, immune responses, and oxidative defenses. This study aimed to assess how a prolonged high-fat diet (HFD) affects the effectiveness of physical activity in preventing and managing mammary tumorigenesis. Ovariectomised C57BL/6 mice were provided with an enriched environment to induce spontaneous physical activity while being fed HFD. After 44 days (short-term, ST HFD) or 88 days (long-term, LT HFD), syngenic EO771 cells were implanted into mammary glands, and tumour growth was monitored until sacrifice. Despite similar physical activity and food intake, the LT HFD group exhibited higher visceral adipose tissue mass and reduced skeletal muscle mass. In the tumour microenvironment, the LT HFD group showed decreased NK cells and TCD8+ cells, with a trend toward increased T regulatory cells, leading to a collapse of the T8/Treg ratio. Additionally, the LT HFD group displayed decreased tumour triglyceride content and altered enzyme activities indicative of oxidative stress. Prolonged exposure to HFD was associated with tumour growth despite elevated physical activity, promoting a tolerogenic tumour microenvironment. Future studies should explore inter-organ exchanges between tumour and tissues.

## 1. Introduction

Breast cancer is the most common cancer diagnosed in women and the second most common for both sexes, just behind lung cancer. In 2022, an estimated 2.3 million new breast cancer cases arose, representing 11.6% of all cancer cases. Breast cancer is the fifth leading cause of cancer death worldwide, with 666,000 deaths. Among women, breast cancer accounts for 1 in 4 cancer cases and 1 in 6 cancer deaths [1]. Breast cancer is a multifactorial pathology. Non-modifiable genetic factors have an important role in mammary carcinogenesis, with the involvement of high and medium penetrant genes [2] and many single nucleotide polymorphisms (SNP) [3], as reported in the literature. These mutations can severely increase the risk of developing breast cancer [4,5]. However, 5–10% of breast cancers are recognised to be hereditary [6]. The contribution of SNP polymorphisms to the occurrence of breast cancer is more closely related to pre-menopausal invasive breast cancer than to post-menopausal cases [7].

The results of numerous clinical and epidemiological studies clearly show the influence of modifiable risk factors such as lifestyle habits on the prevention and development of breast cancer [8]. Indeed, an overall healthy lifestyle is inversely related to the risk of invasive breast cancer, in particular, among post-menopausal women [7,9].

As stated by the World Cancer Research Fund International [10], excess weight or obesity throughout adulthood has an ambivalent role depending on menopausal status. Obesity increases the risk of getting breast cancer by 33% in post-menopausal women, despite being protective in pre-menopausal women [11]. Despite these observations, it is recommended to maintain a healthy throughout all stages of life, with a healthy diet and regular physical activity [10].

Breast tissue is composed of many anatomical structures, including fat, which are involved in various breast mechanisms like development, pregnancy, lactation, and age-related involution [12,13]. The main immune cells found in breast adipocytes are macrophages. These exert various tissular functions, such as removing dead adipocytes and beige adipogenesis, and they contribute to regulating lipid storage [14].

The extent of adipose tissue due to excessive caloric intake and lack of energy expenditure leads to an alteration of white adipose tissue (WAT) physiology throughout the whole body, including in organs such as the breast. This is associated with many pathologies [15], such as inflammation, insulin resistance [16], and metabolic syndrome [17]. Excess calorie consumption during obesity can also trigger liver disease [18], which could then lead to extrahepatic cancers, such as breast cancer [19].

The link between body fat content and breast cancer is based on several mechanisms. Indeed, chronic inflammation, associated with expended mammary WAT, leads to macrophage (M) infiltration, forming crown-like structures around dead and dying adipocytes as well as in visceral and subcutaneous adipose tissue [20]. This is followed by the activation of nuclear factor-kappa B (NF-κB), inducing elevated production of pro-inflammation factors, such as tumour necrosis factor (TNF)-α, interleukin (IL)-1β, IL-6, and prostaglandin E (PGE)2, and dysregulation of the leptin/adiponectin ratio. The increase in leptin level contributes to many mechanisms in breast cancer development, such as cell proliferation, adhesion, invasion, migration, inflammation, angiogenesis, and epithelial–mesenchymal transition [21]. The upregulation of pro-inflammatory factors is associated with an increase in the expression and activity of aromatase in WAT. The final stage of estrogen synthesis is regulated by this enzyme [22], which in turn causes an increase in mitochondrial respiration linked to the generation of reactive oxygen species, DNA damage from mitochondrial metabolism, and a decrease in the DNA damage repair system [23]. All of these phenomena are responsible for the increased risk of breast cancer in obesity.

Physical activity is recommended for overweight or obese people for cancer prevention and survival [24,25]. Such activity reduces body fat mass and obesity-associated inflammation, restores insulin sensitivity, and increases energy expenditure and metabolite consumption [26,27]. Physical activity is also proposed as a solution for the management of obesity-related-liver disease [28]. It acts on many levels against breast cancer. Physical activity decreases the availability of androgens, estrogens, and progestogens and increases the production of sex hormone-binding globulin [29]. Physical activity modulates estrogen signaling associated with breast cancer in obese individuals. At the immune level, we previously demonstrated that, in a high-fat diet-fed mice, spontaneous physical activity permits the recruitment of antitumour immune cells in the tumour microenvironment, leading to a decrease in tumour growth [30]. This observation can be explained by multiple mechanisms, such as the recruitment of CD8^+^ T cells [31] or the modulation of natural killer (NK) cell proportion and activity [32]. Moreover, oxidative stress is reduced with an increase in the expression and/or the activities of antioxidant enzymes, depending on the type of physical exercise [33]. The role of physical activity in metabolic regulation to counteract cancer development is discussed as well [34].

In previous animal studies on post-menopausal breast cancer, we have shown that spontaneous physical activity prevents tumour growth at several levels, such as immune regulation inside the tumour [30], or at systemic and tissular hormone levels [35]. These studies focused on a short-term high-fat diet; therefore, it is unknown how long-term obesity may affect the benefits of physical activity. Consequently, this study aims to evaluate the benefits of spontaneous physical activity on the effect of a long-term high-fat diet (LT HFD) versus a short high-fat diet (ST HFD) duration in mammary tumour development in a syngenic mice model of post-menopausal cancer.

## 2. Results

### 2.1. A Long-Term High-Fat Diet Tended to Influence Spontaneous Physical Activity but Not Food Intake

Before tumour cell implantation, LT HFD had no impact on the daily distance travelled (LT HFD = 1.065 ± 0.076 vs. ST HFD = 0.693 ± 0.103 km/day/mouse, *p* = 0.1265). Similarly, tumour growth did not affect the daily distance travelled, whatever the group (LT HFD = 0.965 ± 0.217 vs. ST HFD = 0.818 ± 0.048 km/day/mouse, *p* = 0.9357). As a whole, the experiment showed an effect of diet duration on spontaneous physical activity that tended to be significant (*p* = 0.0915), contrary to tumour growth (*p* = 0.9335).

Food intake was similar between the two groups, either before or after tumour implantation. Similarly, there was no difference in food intake before and after tumour implantation within each group (Figure 1A).

### 2.2. The Duration of Lipid Consumption Altered Anthropometric Parameters

When the high-fat diet was introduced, both groups significantly gained weight until tumour implantation. During tumour growth, mice under LT HFD did not lose weight (LT HFD = 32.3 ± 1.3 g at sacrifice), while mice under ST HFD tended to lose weight at sacrifice (ST HFD = 28.4 ± 0.8 g, *p* = 0.0856), leading in a significant effect of diet duration on weight management (*p* = 0.0109). It is noteworthy that, in each group, one mouse did not respond to the high-fat diet (LT HFD: 25.3 g, 24.7 g, and 25.5 g; ST HFD: 23.5 g, 25.2 g, and 24.8 g, respectively) at the beginning, implantation, and sacrifice times (Figure 1B,D).

Regarding average weight evolution before tumour implantation, daily weight gain was similar between the two groups (LT HFD = 0.101 ± 0.013 vs. ST HFD = 0.139 ± 0.020 g/day). During tumour growth, mice under LT HFD lost less weight than those under ST HFD (LT HFD = −0.045 ± 0.011 vs. ST HFD = −0.138 ± 0.022 g/day, *p* = 0.0013) (Figure 1C).

Whatever the diet duration, at sacrifice, total absolute adiposity remained the same between groups (Figure 2A) but differed according to the adipose tissue localisation. LT HFD significantly increased visceral adipose tissue weight (LT HFD = 1533 ± 260 vs. ST HFD = 1029 ± 204 mg, *p* = 0.0446) (Figure 2B) and decreased brown adipose tissue (LT HFD = 87 ± 10 vs. ST HFD = 171 ± 19 mg, *p* = 0.0004) (Figure 2C).

The total muscle mass resulting from the sum of skeletal muscle masses of the two hind legs tended to be lower in the LT HFD group in comparison to the ST HFD group (334 ± 8 vs. 354 ± 13 mg, *p* = 0.0765) (Figure 2D). Similarly, in response to LT HFD, the relative total muscle mass decreased significantly (Figure 2E) as a consequence of reduced individual muscle mass, such as the gastrocnemius, (LT HFD = 0.72 ± 0.03% vs. ST HFD = 0.90 ± 0.01%, *p* = 0.0001) (Figure 2F), soleus (LT HFD = 0.06 ± 0.002% vs. ST HFD = 0.07 ± 0.004%, *p* = 0.0544) (Figure 2G), and tibialis (LT HFD = 0.26 ± 0.01% vs. ST HFD = 0.35 ± 0.03%, *p* = 0.0068) (Figure 2H).

### 2.3. Biochemical Profiles in Liver and Plasma

LT HFD did not affect liver weight (LT HFD = 1.38 ± 0.06 vs. ST HFD = 1.36 ± 0.06 g) nor its glucose, triglycerides, cholesterol, urea, and uric acid concentrations. Only the total bilirubin concentration was significantly higher in response to longer lipid exposition (Table 1).

At the circulating level, the concentrations of biomarkers studied were not altered, whatever the diet duration (Table 2).

### 2.4. Tumour Weight and Density

LT HFD induced a significantly higher tumour weight (LT HFD = 2.23 ± 18 vs. ST HFD = 1.81 ± 11 g, *p* = 0.0412) (Figure 3A). However, tumour density remained similar between the two groups (LT HFD = 0.94 ± 0.06 vs. ST HFD = 0.95 ± 0.08 g/cm^3^) (Figure 3B).

### 2.5. Tumour Growth

Tumour growth was faster in response to LT HFD compared to the other group. Indeed, the tumour volume reached 1500 mm^3^ at 21 days post-implantation in LT HFD mice versus 28 days post-implantation in the ST HFD group (Figure 4A). The survival rate was significantly reduced in the LT HFD group (*p* = 0.0296), with median survivals of 22 days for the LT HFD group and 25.5 days for the ST HFD group (Figure 4B).

### 2.6. Tumour Characteristics

#### 2.6.1. Tumour Metabolism

In response to the longest HFD, the tumour glucose concentration tended to increase (LT HFD = 0.159 ± 0.071 vs. ST HFD = 0.061 ± 0.012 mmol, *p* = 0.1, Figure 5A) and the triglyceride content was significantly reduced (LT HFD = 0.031 ± 0.020 vs. ST HFD = 0.114 ± 0.029 mmol, *p* = 0.0571; Figure 5B). The tumour cholesterol level remained unchanged (LT HFD = 0.071 ± 0.036 vs. ST HFD = 0.042 ± 0.007 mmol, Figure 5C).

The expression of genes involved in energy metabolism and more specifically in energy production was modulated by lipid diet duration. While no change was observed in *carnitine palmitoyl transferase (Cpt) 2* gene expression (LT HFD = 0.95 ± 0.05 vs. ST HFD = 1.01 ± 0.08, Figure 5E), the expression of *Cpt1* tended to increase in response to the longest lipid impregnation (LT HFD = 2.05 ± 0.21 vs. ST HFD = 1.11 ± 0.24, *p* = 0.1111, Figure 5D). The gene coding for *citrate synthase* was significantly overexpressed (LT HFD = 1.38 ± 0.09 vs. ST HFD = 1.00 ± 0.04, *p* = 0.0159, Figure 5F).

#### 2.6.2. Tumour Oxidative Stress

Among antioxidant enzymes, the total tissular activity of glutathione peroxidase (GPx) (LT HFD = 15.33 ± 1.11 vs. St HFD = 3.46 ± 1.35 UI, Figure 6A) and the expression of the gene coding for the isoform *Gpx1* (LT HFD = 1.58 ± 0.13 vs. ST HFD = 1.04 ± 0.15, *p* = 0.0159, Figure 6B) and the isoform *Gpx2* (LT HFD = 2.03 ± 0.24 vs. ST HFD = 1.04 ± 0.13, *p* = 0.0040, Figure 6C) were significantly increased in response to LT HFD.

Tumour superoxide dismutase (SOD) activity tended to be higher in response to the longest lipid exposure (LT HFD = 2387 ± 708 vs. ST HFD = 1202 ± 189 UI, *p* = 0.1143, Figure 6D). At the transcriptomic level, the expression of *Sod1* (LT HFD = 1.46 ± 0.11 vs. ST HFD = 1.03 ± 0.15, *p* = 0.0476, Figure 6E) was significantly increased in the LT HFD condition while the expression of *Sod2* remained unchanged (LT HFD = 1.07 ± 0.09 vs. ST HFD = 1.08 ± 0.24, Figure 6F).

In response to LT HFD, tumour reduced glutathione (GSH) content rose significantly (LT HFD = 0.015 ± 0.002 vs. ST HFD = 0.007 ± 0.001 mmol, *p* = 0.1, Figure 6G) while *glutamate-cysteine ligase catalytic subunit* (*Gclc*) expression was significantly reduced (LT HFD = 0.80 ± 0.06, *p* = 0.0476 vs. ST HFD = 1.04 ± 0.14, Figure 6H). Glutathione reductase (GR) activity tended to be enhanced (LT HFD = 41 ± 10 vs. ST HFD = 16 ± 2 UI, *p* = 0.1, Figure 6I) and *glutathione-disulfide reductase* (*Gsr*) expression was significantly upregulated (LT HFD = 1.61 ± 0.15 vs. ST HFD = 1.05 ± 0.19, *p* = 0.0278, Figure 6J) in response to LT HFD.

The activity of glutathione S transferase (GST) was significantly reduced in the LT HFD group (LT HFD = 6.7 ± 2.5 vs. ST HFD = 30.9 ± 4.3 UI, *p* = 0.0143, Figure 6K), whereas the expression of *glutathione S-transferase omega* (*Gsto*) (LT HFD = 1.74 ± 0.16 vs. ST HFD = 1.01 ± 0.06, *p* = 0.0040, Figure 6L) was enhanced.

#### 2.6.3. Tumour Immune Infiltrate

In the tumour microenvironment, following LT HFD exposition, a significant reduction in the proportion of both NK cells (LT HFD = 0.2 ± 0.1 vs. ST HFD = 24.5 ± 1.9%/total T lymphocytes, *p* = 0.05, Figure 7A) and T8 cells (LT HFD = 1.008 ± 0.275 vs. ST HFD = 7.875 ± 3.068%/total T lymphocytes, *p* = 0.0022, Figure 7B) was observed, while the Treg cell proportion tended to increase (LT HFD = 0.5 ± 0.06 vs. ST HFD = 0.017 ± 0.007%/ lymphocyte T, *p* = 0.1, Figure 7C). Consequently, the T8/Treg ratio, a marker of antitumoural immune defense, fell (LT HFD = 2.6 ± 1.3 vs. ST HFD = 421.6 ± 153.8, *p* = 0.0286, Figure 7D).

Diet duration did not affect the global tumour-associated macrophage proportion (LT HFD = 66.7 ± 6.8 vs. ST HFD = 60.3 ± 10.7%/total leucocytes, Figure 7E). At the transcriptomic level, LT HFD was linked to non-significant downregulated expression of *nitric oxide synthase* (*Nos) 2*, a marker of M1 (LT HFD = 0.71 ± 0.35 vs. ST HFD = 1.05 ± 0.18, *p* = 0.1111, Figure 7F), whereas the expression of *mannose receptor c-type (Mrc) 1*, a marker of M2, remained stable (LT HFD = 0.86 ± 0.31 vs. ST HFD = 1.15 ±0.27, Figure 7G).

The T lymphocyte helper (Th) 1 cell proportion increased in the LT HFD group (LT HFD = 0.203 ± 0.088 vs. ST HFD = 0.068 ± 0.016%/ T lymphocytes, *p* = 0.0595, Figure 7H) without a change in Th2 cell infiltrate (LT HFD = 8.530 ± 6.6 vs. ST HFD = 8.435 ± 1.673%/T lymphocytes, Figure 7I). Thus, the Th1/Th2 ratio, a marker of antitumoural immune defense, tended to increase (LT HFD = 0.024 ± 0.013 vs. ST HFD = 0.011 ± 0.004, *p* = 0.0714, Figure 7J).

## 3. Discussion

This animal model of aged ovariectomised mice fed a hypercaloric diet, mimicking menopause and the risk of developing breast cancer, has been previously characterised and validated [30,35,36]. The EO771 tumour cells represent a luminal B cancer type [37,38] and are particularly immunogenic [30].

In this model, we previously observed that a 3-month HFD led to an increase in body fat and maintenance of muscle mass compared to a normocaloric diet. HFD was also associated with rapid tumour growth and a higher final tumour mass. Spontaneous physical activity was reduced under HFD and tumour bearing (Appendix A) [39]. With physical activity being key to the prevention of obesity and breast cancer [34], the animals were housed in an enriched environment known to increase not only spontaneous physical activity but also animal well-being [40]. Indeed, previous studies have shown that spontaneous physical activity promoted by housing in an enriched environment provides a protective factor against cancer and limits tumour growth via hormonal, cellular, and immune mechanisms [30,35,40,41]. It is well known that weight gain in any adult’s life period is associated with increased postmenopausal breast cancer risk in humans [42]. Thus, based on our experimental model, the objective of this study was to see if the beneficial effects of spontaneous physical activity on tumour growth were maintained in response to a long high-fat diet duration.

Both HFD groups were housed in the same enriched environment, allowing an equivalent level of spontaneous physical activity. The ST HFD lasted 44 days and the LT HFD lasted twice that time (88 days), at tumour implantation. Throughout the experiment, independently of food intake, body weight changed. Indeed, the body weights of the two groups of mice increased in the same range throughout the experiment, even though the LT HFD group was fed for a longer time. This could be explained by the fact that, in parallel with the longer diet time, there was also longer physical activity practice, which is known to regulate body weight [26]. The daily gain in mass of the LT HFD group was lower but statistically similar to that of the ST HFD group, which is why animals from both groups reached the same body weight before the time of tumour implantation.

Despite similarities in weight gain in the pre-tumoural period, after tumour implantation, the ST HFD group lost more weight than the LT HFD group. This difference led to corporal changes at sacrifice. In our experiment, the body weight loss observed in the short-term group was not intended but was associated with lower tumour growth. This body weight loss seemed to be supported mainly by a lower visceral adipose tissue mass. This observation is in accordance with the literature reporting the beneficial effects of physical activity, not only on prevention but also on the therapeutic management of breast cancer [42,43]. Conversely, mice under LT HFD presented more visceral adipose tissue, less total hind leg muscle mass, and higher tumour growth. These data are in line with recent studies reporting that muscle loss may adversely affect breast cancer patient outcomes, independently from malnutrition [44]. Moreover, the muscle mass variation observed could be an interesting perspective to follow the low but real prevalence of sarcopenia in women with breast cancer [45].

The absence of differences between the two groups of mice in plasma and liver biochemical markers could be explained by the time course of metabolic alteration appearance. Indeed, at sacrifice, the LT HFD and ST HFD groups were respectively at 110 and 70 days of the high-fat diet. It has been reported in male mice that the first plasma biological changes leading to metabolic syndrome appear only after at least 90 days of a high-fat diet, and the major perturbations are observed at 150 days compared to a standard diet [46]. Moreover, focusing on the high-fat diet group, all of the metabolic parameters were similar between 60, 90, and 120 days. Thus, the lack of differences in liver and plasma markers in our model could be explained by the early metabolic time window, whatever the diet duration.

Surprisingly, tumour growth was influenced by the duration of the diet. Indeed, tumorigenesis was faster in the LT HFD group, leading to a reduced survival rate. The long-term high-fat diet decreased the tumour triglyceride content while the glucose level tended to increase. Similarly, tumour overexpression of *Cpt1* and *Cs* supports reprogramming of energy metabolism in favour of fatty acid oxidation (FAO). As is known, the energetic pathway of cancer cells mostly relies on glycolysis (Warburg effect) [47]. However, in an adipocyte-rich microenvironment, breast cancer cells may switch from typically glucose-centered to lipid-centered metabolism. This lipid-centered connection is particularly pronounced in obesity [48].

Lipid oxidation in the mitochondria leads to oxidative stress, which is associated with the production of reactive oxygen species. This latter is well known to favour the development of breast cancer [49] and be differently regulated in cancer cells under obesogenic conditions [50]. In our model, the tumour increase in antioxidant defense markers under LT HFD, such as SOD, the glutathione recycling enzyme GR, the lipid peroxidation protecting GPx, and the reduced GSH, could be linked to higher metabolic oxidative stress due to the activation of FAO [51]. Moreover, this fatty acid metabolism pathway has been demonstrated to protect cancer cells against apoptosis. Increased FAO triggers the activation of signal transducer and activator of transcription 3 (STAT3) through acetylation, which enhances mitochondrial integrity and reduces apoptosis [52]. These mechanisms are in accordance with the SU.VI.MAX cohort study’s results showing that oxidative stress [53] and lipid metabolites [54] are markers of long-term breast cancer risk.

Oxidative stress and lipid peroxidation are major contributors to inflammation in the tumour. Inflammation is one of the hallmarks of cancer and is a key regulator of tumour immune infiltration [55]. Inflammatory molecules recruit immune cells in the tumour microenvironment [56]. In our experiment, the long-term high-fat diet led to an increase in the proportion of Treg cells and a decrease in CD8^+^ T cells and NK cells despite elevated spontaneous physical activity. Moreover, the long-term diet tended to decrease the expression of M1 marker, without affecting the M2 marker. In our experimental conditions, LT HFD induced a pro-tumoural immune infiltrate. Indeed, the switch in the polarisation of macrophages from pro-inflammatory M1 to anti-inflammatory M2 [57] is known to favour the recruitment of T regulator (Treg) lymphocytes [58]. All of these cells together inhibit antitumoural immunity indirectly by modulating T helper (Th) lymphocytes, favoring Th2 instead of Th1 lymphocytes, but also directly by inhibiting the cytotoxic T cells CD8^+^ and NK cells responsible for the destruction of tumour cells [59]. Associated with the long-term high-fat diet, the elevated spontaneous physical activity was not able to maintain an efficient antitumour immune defense, unlike in our previous observations [30]. Altogether, these results led to a tolerogenic environment, favourable to the development of mammary cancer.

Our experiment is one of the few studies in the literature performed on old ovariectomised female mice. The main limitation of our study is the fact that the two groups are in the same window for metabolic syndrome development [46]. A longer HFD duration should be tested to develop a more severe metabolic syndrome and highlight some other mechanisms that could be implicated, such as liver alteration and higher insulin resistance. Moreover, in agreement with the ethical 3R rules, we did not include a standard diet control group, as was previously included (Appendix A).

However, this model, close to physiopathological breast carcinogenesis, demonstrates the celerity of a high-fat diet in promoting tumour growth in association with a switch in the immune population. This chain of events seems to happen even without plasma metabolic disturbances and in the presence of spontaneous physical activity. To our knowledge, few papers have described a spontaneous loss of muscle in mouse models of breast cancer. Further investigations could be relevant to understand the relationship between the duration of the high-fat diet and the modulation of body mass, especially the increase in visceral adipose tissue gain and in muscle mass loss. The model with a long-term high-fat diet will be used to explore the inter-organ dialogue in terms of cytokines, adipokines, hepatokines, and exerkines [60,61,62,63].

## 4. Materials and Methods

### 4.1. Animal Model and Housing Conditions

This experiment was carried out in accordance with European directives on animal ethics (Comité Régional d’Ethique sur l’Expérimentation Animale, No. 01095.02, Clermont-Ferrand, France). Female mice C57BL/6J, 33-weeks-old, purchased from Janvier Lab (Le Genest-St-Isle, France), were placed in a box under standard laboratory conditions (i.e., 12 h light and 12 h dark cycle on a reverse light cycle and 22 ± 2 °C). Diet and water were accessed ad libitum. The cages were enriched with toys, a nest, and a wheel, with 5–7 mice per cage. This enriched environment permitted the promotion of spontaneous physical activity, which is well known to enhance social interaction due to multiple mice per cage, to influence normal mammary gland development, and to inhibit mammary tumour growth [41]. All mice were ovariectomised to mimic the menopausal effect, which is absent in the species.

### 4.2. Diet, Body Weight Follow Up and Physical Activity

All mice were fed a high-fat (HF) diet designed in collaboration with the laboratory “Safe” (Augy, France), following the American Institute of Nutrition 93 (AIN-93G) recommendations, and previously published [30]. With 450 kcal/100 g of food, it is composed of 42.1% lipids (ratio ω6/ω3 = 6.2; ω6 intake: 2.46 mg/100 g), 37.4% carbohydrates, and 20.5% protein. All minerals and vitamins were present in optimal quantities. Body weight and ad libitum food intake were measured twice a week throughout the experimental period. Spontaneous physical activity was assessed using wheel counters in each cage or using a TSE System PhenoMaster/LabMaster (TSE System, Bad Homburg, Germany), depending on the difficulties encountered during the experiments. With the wheel counter, the distance travelled per cage, relative to the number of mice, was recorded daily. For the TSE system, the spontaneous physical activity was measured twice during the experiment, before and after tumour implantation, with 1 day of acclimatisation and 3 days of measurements each time.

### 4.3. Orthotopic Injection of EO771 Mammary Adenocarcinoma Cells and Tumour Monitoring

The C57BL/6 syngeneic cell line of spontaneous mammary adenocarcinoma EO771 was cultured in complete RPMI 1640 medium (Biowest, Nouaille, France) supplemented with 10% foetal calf serum (Biowest), 100 μg/mL of streptomycin (Sigma-Aldrich, Saint-Quentin Fallavier, France), 100 U/mL of penicillin (Sigma-Aldrich), and 2 mM glutamine (Sigma-Aldrich) at 37 °C in a 5% CO_2_ humidified atmosphere. Tumour cells were implanted into the mouse’s left fourth pair of mammary glands using the “fat pad” technique, with a density of 5 × 10^5^ cells/100 μL of matrigel (Growth Factor Reduced BD Matrigel TM Matrix, BD Biosciences, Bedford, MA, USA). The surgery was performed after 44 days of HF diet for the short-term high-fat diet group (ST HFD, *n* = 10) and 88 days for the long-term high-fat diet group (LT HFD, *n* = 11). The tumour size was determined three times a week by measuring the perpendicular diameter with a digital calliper. The tumour volume was calculated using the formula V = 4π/3 × (width/2)^2^ × (length/2), where the width is the smaller of the two measurements.

### 4.4. Sacrifice and Blood and Tissue Sampling

The experiment ended when the tumour volume exceeded 2 cm^3^, animal weight loss of more than 20%, or at a maximum of 35 days post-implantation. The mice were anaesthetised with a ketamine/xylazine mixture (i.p., 100/10 mg/kg) and blood was collected by cardiac puncture in heparinised tubes. After centrifugation at 1000× *g* for 5 min, plasma was collected, and aliquots were made and stored at −80 °C. Several organs were harvested and weighed, then frozen in liquid nitrogen, and stored at −80 °C, notably, the different adipose tissues, the gastrocnemius, soleus, and tibialis skeletal muscles, and the liver. The tumours were collected and cut into pieces, with one piece used for the study of immune cell infiltration and the others frozen.

### 4.5. Tumour Immune Cell Infiltration

The tumour piece analysed, after collection at sacrifice, was crushed in sterile phosphate-buffered saline (PBS, Biowest) 1X with 0.5% bovine serum albumin (BSA, Sigma-Aldrich), filtered using a 40 μm pore filter (Falcon^®^ 40 μm Cell Strainer), centrifuged to pellet tissue debris, and then resuspended in staining buffer (PBS 1×, BSA 0.5%, ethylene diamine tetra acetic acid (EDTA, Fluka, Sigma-Aldrich) 2 mM). According to the supplier’s recommendations (MACS Miltenyi Biotec, Bergisch Gladbach, Germany or eBioscience, San Diego, CA, USA), immune cells were labelled by monoclonal antibodies conjugated to fluorochromes (Table 3). Cells were surface-labelled with specific antibodies for 30 min at 4 °C. Intracellular staining using anti-FoxP3-Biotin was performed according to the manufacturer’s instructions (eBioscience). The samples were then analysed by flow cytometry (Beckman Coulter EPICS XL FC500, Villepinte, France), where immune populations were characterised and counted based on their size and granularity. Due to the constraints of the methods, half of the animals were randomised and used in cytometric analysis.

The data were post-analysed using Kaluza 1.2 software (Beckman-Coulter, Hialeah, FL, USA).

### 4.6. Tissue Exploration

#### 4.6.1. Tissue Preparation

Tumours and livers were cut into pieces before homogenisation using an Ultraturax^®^ system in 100 mM Tris, 1 mM EDTA, supplemented with protease (Halt Protease Inhibitor Cocktail, Thermo Scientific, Waltham, MA, USA) and phosphatase (Halt Phosphatase Inhibitor Cocktail, Thermo Scientific) inhibitors for metabolic and protein exploration or in TRIzol^®^ reagent (Invitrogen, Saint Aubin, France) for RNA exploration. The samples were homogenised and centrifuged at 500× *g* for 5 min. The supernatant was aliquoted and frozen until analysis at −80 °C.

#### 4.6.2. RNA Extraction

Total tumour RNA was isolated using TRIzol^®^ reagent according to the manufacturer’s instructions and quantified using a Spark^®^ multimode microplate reader (Tecan Trading AG, Männedorf, Switzerland). Reverse transcription was carried out in a thermocycler (Mastercycler^®^ gradient; Eppendorf, Montesson, France) on 1 μg of total RNA for each condition using a high-capacity cDNA reverse transcription kit (Applied Biosystems, Saint Aubin, France) with random hexamer pdN6 primers.

#### 4.6.3. Quantitative Real-Time PCR

According to the manufacturer’s protocol, qPCR was carried out using SYBR^®^ Green reagent on a StepOne system (Applied Biosystems, Waltham, MA, USA). Each sample was assayed in duplicate. Relative quantification was obtained by the comparative Ct method, based on the formula 2^−ΔΔCt^. The expression level of genes involved in energetic metabolism was normalised to that of the housekeeping gene *Hprt*. The fold-change of expression was determined against the ST HFD condition set at level one. The sequences and fragment sizes of the murine-specific primers used are reported in Table 4.

#### 4.6.4. Protein Quantification

Protein quantities were assayed using a bicinchoninic acid (BCA) assay (Interchim, Montluçon, France) based on the Biuret method with a microplate spectrophotometer reader at 550 nm (Multiskan FC, Thermo Scientific, Waltham, MA, USA).

#### 4.6.5. Enzymatic Activities

All enzymatic activities were analysed using the Thermo Electron Konelab 20i^®^ (Thermo Scientific, Waltham, MA, USA). All total tissular enzyme activities are expressed in UI.

The glutathione reductase assay was based on a two-reagent method. Reagent 1 was composed of 100 mM of Tris, 1 mM of EDTA, and 87 µM of NADPH, H^+^, with the pH adjusted at 8. Reagent 2 was the result of adding 0.1 mM of glutathione disulfide (GSSG) to reagent 1. Thirty µL of sample was incubated with 200 µL of reagent 1 and 10 µL of reagent 2. The assay was performed at 37 °C by measuring the absorbance at 340 nm every 30 s for 3 min. For quantification, the enzymatic factor used was −1270.

The glutathione peroxidase assay was based on a one-reagent method. The reagent was composed of 100 mM, 1 mM of EDTA, 0.156 mM of NADPH, H^+^, 0.1 UI/mL of glutathione reductase, 5 mM of glutathione (GSH), and 22.2 mM of tert-butyl hydroperoxide (tBOOH), with the pH adjusted at 7.4. Thirty µL of sample was added to 210 µL of reagent, and the enzyme activity was studied by measuring the absorbance at 340 nm every 30 s for 5 min at 37 °C. For quantification, the enzymatic factor used was −1270.

The glutathione S-transferase assay was based on a two-reagent method. Reagent 1 was made by mixing 1 mL of 50 mM of GSH with 7 mL of 50 mM 4-(2-hydroxyethyl)-1-piperazineethanesulfonic acid (HEPES) and Triton 0.1X, with the pH adjusted at 7.5. Reagent 2 was composed of 1 mM of 1-chloro-2,4-dinitrobenzene (CDNB) in 40% ethanol. After mixing 20 µL of sample with 160 µL of reagent 1 and 20 µL of reagent 2, the assay was performed at 37 °C by measuring the absorbance at 340 nm every 30 s for 3 min. For quantification, the enzymatic factor used was 1042.

Superoxide dismutase (SOD) activity was measured using an SOD activity kit (#MAK379-1KT, Sigma-Aldrich). Briefly, 20 µL of sample or standard was mixed with 200 µL of WST working solution and 20 µL of enzyme working solution, and the absorbance was read at 0 and 20 min. Quantification of enzymatic activity was obtained with a calibration curve according to the manufacturer’s instructions.

#### 4.6.6. GSH Assay

The reduced gluthatione (GSH) assay was based on the reaction with 5,5′-dithiobis-2-nitrobenzoic acid (DTNB). The DTNB reagent was composed of 2 mM of DTNB in 50 mM K_2_HPO_4_ buffer solution. Multiple standard concentrations (5 mM solution used at purity and 1/2, 1/4, 1/8, 1/16, 1/32, and 1/64 dilutions) were used to obtain a standard range. For the reaction, 10 µL of standards and sample was mixed with 200 µL of DTNB reagent, and the absorbance was read at 405 nm after 3 min of reaction.

### 4.7. Metabolism Assay

The concentrations of glucose (ref #80009), triglycerides (ref #80019), cholesterol (ref #80106), total bilirubin (ref #80443), urea (ref #80221), uric acid (ref #80351), and creatinine (ref #80107) were quantified using suitable kits (Biolabo, Maizy, France) based on colourimetric measurements. The concentrations of glucose, triglycerides, and cholesterol were measured in the plasma, and bothtumour, and liver supernatants. The levels of total bilirubin, uric acid, urea, and creatinine were evaluated in the plasma and liver supernatants.

### 4.8. Statistical Analysis

All results are expressed as the mean ± SEM and were analysed statistically with Prism 8.0.2 software for Windows (GraphPad Software Inc., San Diego, CA, USA). Comparisons between the two groups were evaluated by the Mann–Whitney test for one variable and two-way analysis of variance (ANOVA) followed by the Tukey test for two variables. The difference in the survival of the animals was assessed using a log-rank Mantel–Cox test. *p*-values < 0.05 indicate a significant difference.

## 5. Conclusions

In conclusion, our study has demonstrated that a longer high-fat diet period prior to breast cancer apparition, associated with elevated spontaneous physical activity, favours tumorigenesis. One of the key mechanisms involved is the immune modulation of the tumour microenvironment. Moreover, these effects appear without the canonical biological markers of metabolic syndrome. Our results highlight the necessity to take charge of obesity and eating disorders as soon as possible in life, and not just take care of the patient when the disease appears. As diet duration could counteract the beneficial effect of spontaneous physical activity, further work is needed to explain all of the mechanisms involved either in the tumour or in other organs like muscle and adipose tissue.

## Figures and Tables

**Figure 1 ijms-25-06221-f001:**
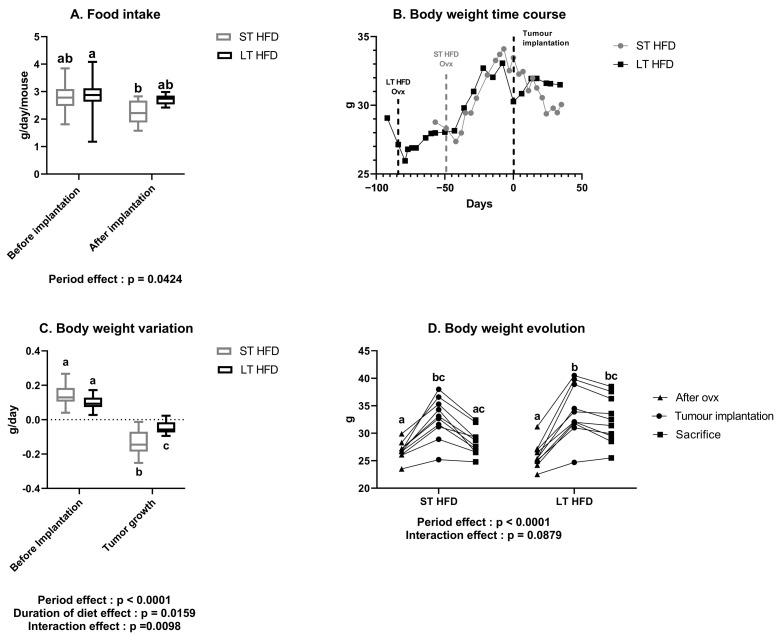
Impact of long-term high-fat diet on daily food intake and body weight evolution. (**A**) Food intake and (**B**) body weight time course throughout the experiment. (**C**) Body weight variation before and after tumour implantation. (**D**) Individual body weight evolution after ovariectomy (ovx) at tumour implantation and sacrifice. Data presented as boxes with median, interquartile range and min–max values (*n* = 10–11 mice/group) or individually were analysed by two-way ANOVA followed by a Tukey test with the factors being the experimentation period and the diet duration. Values with different superscript letters are statistically different, *p* < 0.05. ST and LT HFD: short- or long-term high-fat diet.

**Figure 2 ijms-25-06221-f002:**
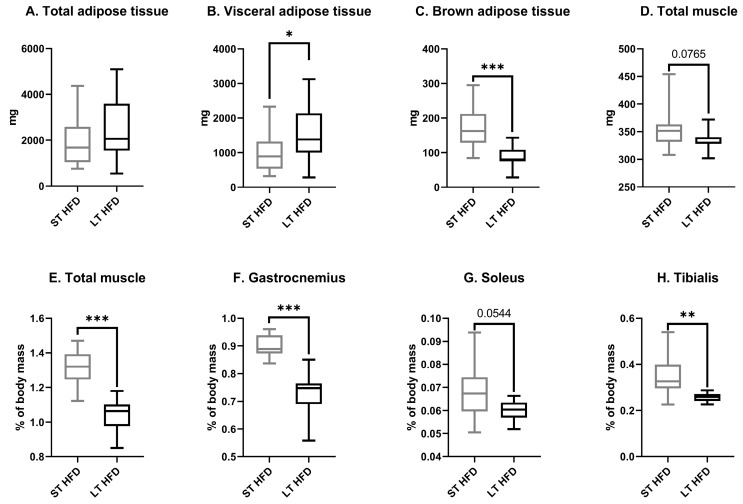
Adipose and muscle masses at sacrifice. Absolute (**A**–**D**) or relative mass (**E**–**H**). Data presented as boxes with median, interquartile range and min–max values (*n* = 10–13 mice/group) were analysed by a Mann–Whitney test, * *p* < 0.05, ** *p* < 0.01, *** *p* < 0.001. ST and LT HFD: short- or long-term high-fat diet.

**Figure 3 ijms-25-06221-f003:**
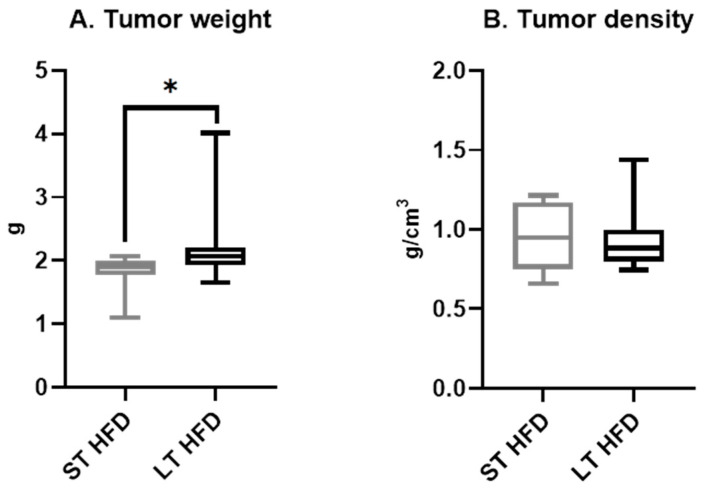
Tumour characteristics. Tumor weight (**A**) and density (**B**) (*n* = 10/13 /group) are presented as boxes with median, interquartile range and min–max values and were analysed by a Mann–Whitney test, * *p* < 0.05. ST and LT HFD: short- or long-term high-fat diet.

**Figure 4 ijms-25-06221-f004:**
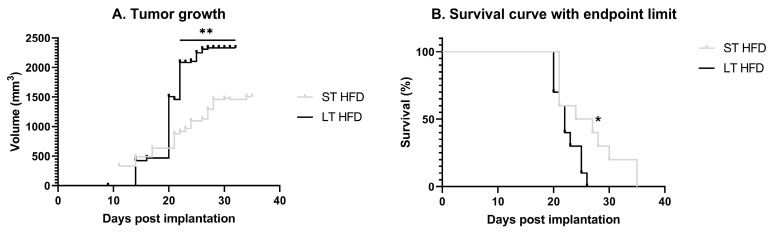
Tumour progression and animal survival. (**A**) Tumour growth evolution depending on the diet duration, results are in volume (mm^3^) with an individual calliper measure. (**B**). Time course of survival with end-point limit. The end-point limit was a 2 cm^3^ tumour, as required by ethical guidelines. Mean ± SEM (*n* = 10–13/group). Data were analysed by repeated measures ANOVA or by Mantel–Cox text as appropriate, * *p* < 0.05, ** *p* < 0.01. ST and LT HFD: short- or long-term high-fat diet.

**Figure 5 ijms-25-06221-f005:**
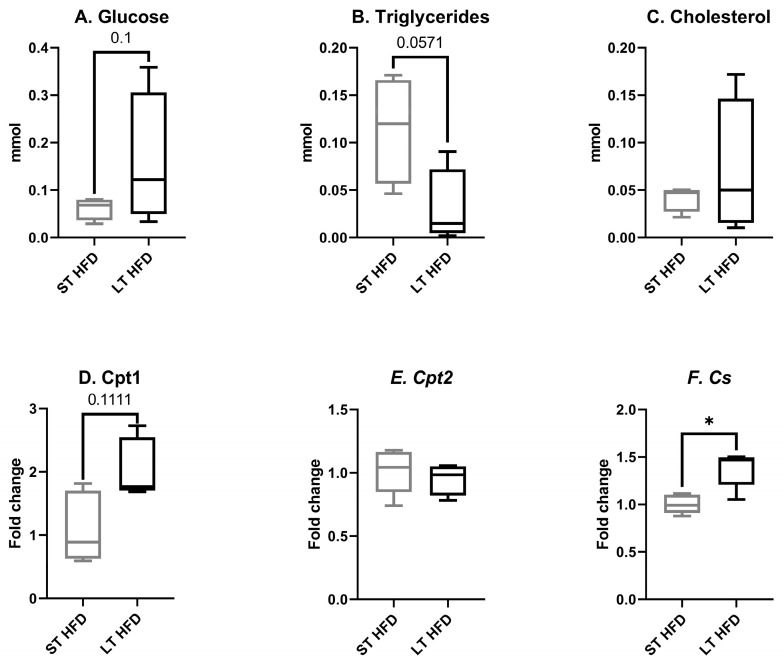
Tumour biological markers and expression of enzymes involved in energy metabolism. (**A**–**C**) Glucose, triglyceride, and cholesterol concentrations were measured using specific kits from Biolabo. (**D**–**F**) *Cpt1*, *Cpt2*, and *Cs* mRNA expression were measured by RT-qPCR and normalised with *Hprt*. Biochemistry assays were performed on 4 mice/group and mRNA expression analysis on 5 mice/group. Results are expressed in min–max ± SEM. Data were analysed by a Mann–Whitney test, * *p* < 0.05. ST and LT HFD: short- or long-term high-fat diet.

**Figure 6 ijms-25-06221-f006:**
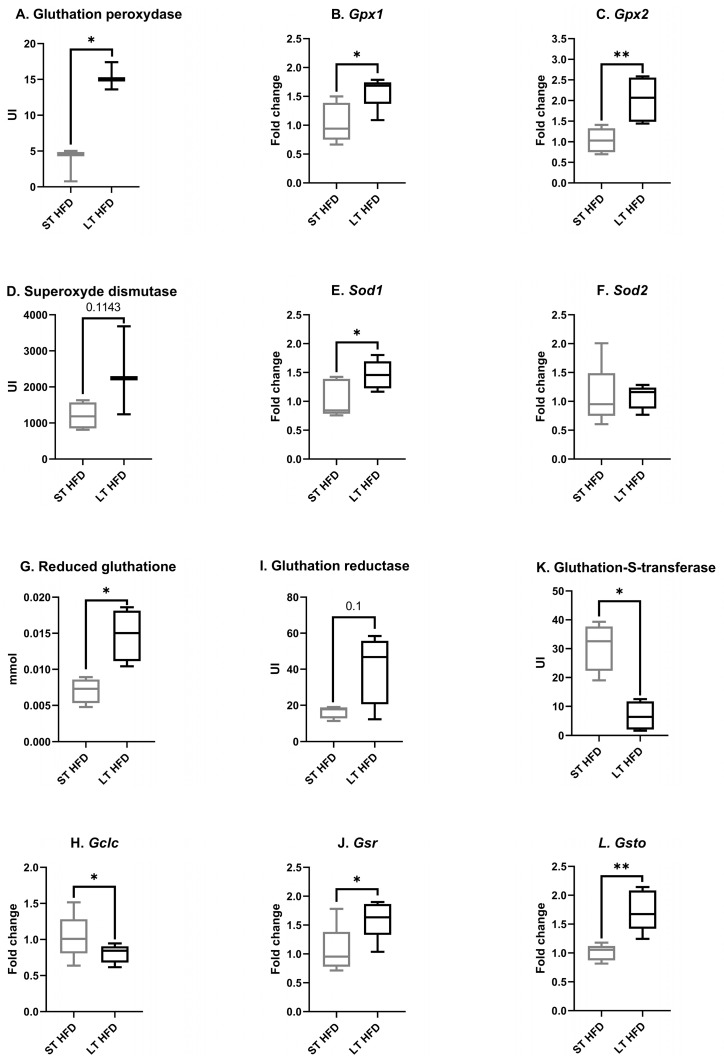
Antioxidant defence in the tumour microenvironment. (**A**,**I**,**K**) Enzyme activities were evaluated using a spectrophotometric assay measuring the disappearance or apparition of the co-substrate NADPH, H^+^. (**D**) Superoxide dismutase (SOD) activity was measured using a kit from Sigma-Aldrich. (**G**) Reduced glutathione (GSH) was quantified using a spectrophotometric assay measuring the appearance of the 2-nitro-5-thiobenzoate (TNB) after the reaction between GSH and 5,5′-dithiobis-TNB. (**B**,**C**,**E**,**F**,**H**,**J**,**L**) mRNA expression was determined by RT-qPCR and normalised with *Hprt*. Enzyme activity assays were performed on 4 mice/group and targeted transcriptomic analysis on 5 mice/group. Results are presented as boxes with median, interquartile range and min–max values. Data were analysed by a Mann–Whitney test, * *p* < 0.05, ** *p* < 0.01. ST and LT HFD: short- or long-term high-fat diet.

**Figure 7 ijms-25-06221-f007:**
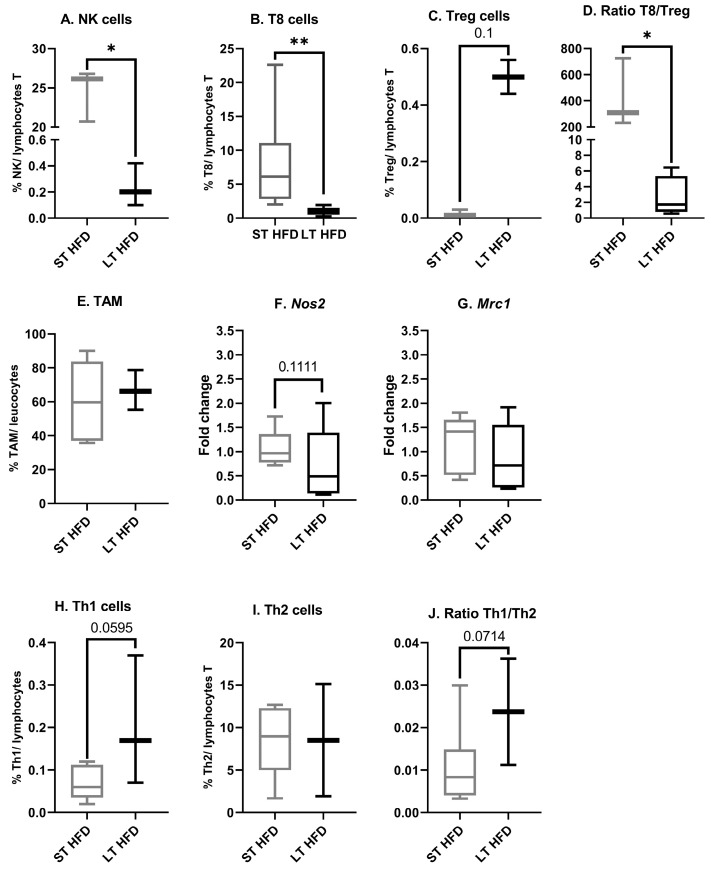
Tumour microenvironnement immune infiltration. (**A**–**E**,**H**–**J**) Sub-populations of immune cells were quantified by flow cytometry using marked antibodies against specific cell surface markers. (**F**,**G**) mRNA expression of *Nos2*, a specific marker of M1, and *Mrc1*, a specific marker of M2, were explored using RT-qPCR and normalised with *Cd45*. Flow cytometry was performed on 10–13 mice/group and mRNA expression analysis on 5 mice/group. Results are presented as boxes with median, interquartile range and min–max values. Data were analysed by a Mann–Whitney test, * *p* < 0.05, ** *p* < 0.01. ST and LT HFD: short- or long-term high-fat diet.

**Table 1 ijms-25-06221-t001:** Liver biochemical markers.

Mmol/Tissue	ST HFD	LT HFD
Glucose	0.045 ± 0.011	0.033 ± 0.006
Triglycerides	0.031 ± 0.011	0.024 ± 0.003
Cholesterol	0.007 ± 0.0003	0.007 ± 0.001
Total bilirubin	0.062 ± 0.018	0.099 ± 0.006 *
Urea	1.61 ± 0.524	1.82 ± 0.099
Uric acid	2.72 ± 0.498	2.50 ± 1.422
Creatinine	0.129 ± 0.043	0.324 ± 0.068

Variables were determined using specific kits from Biolabo. Data are presented as mean ± SEM (*n* = 4/group) and were analysed by a Mann–Whitney test, * *p* < 0.05 vs. ST HFD. ST and LT HFD: short- or long-term high-fat diet.

**Table 2 ijms-25-06221-t002:** Plasma biological markers.

Biological Marker	ST HFD	LT HFD
Glucose (mmol/L)	15.1 ± 5.2	10.8 ± 3.0
Triglycerides (mmol/L)	1.05 ± 0.60	1.11 ± 0.08
Cholesterol (mmol/L)	2.95 ± 0.18	5.65 ± 0.23
Total bilirubin (µmol/L)	8.94 ± 0.39	7.10 ± 1.06
Urea (mmol/L)	12.1 ± 3.9	7.6 ± 0.6
Uric acid (µmol/L)	208 ± 48	142 ± 33
Creatinine (µmol/L)	0.104 ± 0.042	0.042 ± 0.018

Variables were measured using specific kits from Biolabo. Results are expressed as mean ± SEM (*n* = 4/group). Data were analysed by a Mann–Whitney test, *p* > 0.05. ST and LT HFD: short- or long-term high-fat diet.

**Table 3 ijms-25-06221-t003:** Antibodies used for flow cytometry staining.

Cell Type	Antibody	Fluorochrome	Clone	Provider
TAM	Cd11b	PerCP Cy5.5	M1/70.15	eBioscience
	F4/80	PE	REA126	MACS Miltenyi
Natural Killer	NK1.1	PerCP Vio^®^700	PK136	MACS Miltenyi
	CD49b (DX5)	PE	DX5	MACS Miltenyi
T4	CD4	PE	GK1.5	MACS Miltenyi
T8	CD8α	FITC	53-6.7	MACS Miltenyi
Th1	CD4	AF700	GK1.5	eBioscience
	CD119	PE	REA189	MACS Miltenyi
Th2	CD4	AF700	GK1.5	eBioscience
	CD124	PE	REA235	MACS Miltenyi
Treg	CD25	PE	PC61.5	eBioscience
	FoxP3	Biotin	FJK-16s	eBioscience

TAM: Tumour-associated macrophage, NK: natural killer, T4: T4 lymphocyte, T8: T8 lymphocyte, Th1: T helper 1 lymphocyte, Th2: helper 2 T lymphocyte, Treg; regulatory T lymphocyte, CD: cluster of differentiation, FoxP3: forkhead box protein 3; PerCP: peridinin chlorophyll protein, Cy: cyanine, PE: phycoerythrine, FITC: fluorescein isothiocyanate, AF700: Alexa Fluor 700.

**Table 4 ijms-25-06221-t004:** PCR Primers.

Gene	Species	Sequence Reference	Amplicon Size	Forward Primer	Reverse Primer
*Hprt*	mouse	NM_013556.2	176	GTAATGATCAGTCAACGGGGGAC	CCAGCAAGCTTGCAACCTTAACCA
*Cd45*	mouse	NM_001111316.2	158	GCTGATGGATGTGGAGCCAA	TCTGATTGTGGGGCTTTCGG
*Cpt1*	mouse	NM_013495.2	336	CTGGCTCTACCATGACGGGA	ATGGACTTGTCAAACCACCTGTC
*Cpt2*	mouse	NM_009949	241	CATCGTACCCACCATGCACT	ATCAAACCAGGGGCCTGAGA
*Cs*	mouse	NM_026444.4	133	CAGCAGTATCGGAGCCATTGAC	ACCACCCTCATGGTCACTATGGAT
*Gclc*	mouse	NM_010295.2	375	TCGACCTGACCATCGATAAGGA	TCATGTTCTCGTCAACCTTGG
*Gpx 1*	mouse	NM_008160.6	150	GCTCATTGAGAATGTCGCGT	TCATTCTTGCCATTCTCCTGGT
*Gpx 2*	mouse	NM_030677.2	108	CGGGACTACAACCAGCTCAA	CTCGTTCTGACAGTTCTCCTGA
*Gsr*	mouse	NM_010344.4	177	GGCACTTGCGTGAATGTTGG	ATAGATGGTGTTCAGGCGGC
*Gsto1*	mouse	NM_010362.3	168	CTCCGAACCTAAGGGAAGCG	TGTGGGCTAGACACTCCTTG
*Sod1*	mouse	NM_011434.1	136	GGAACCATCCACTTCGAGCA	CTGCACTGGTACAGCCTTGT
*Sod2*	mouse	NM_013671.3	179	GAACAATCTCAACGCCACCG	CCAGCAACTCTCCTTTGGGTT
*Nos2*	mouse	NM_010927.4	131	AGGGTCACAACTTTACAGGGAG	GTGAGGAGCCTCAGAAGTGTC
*Mrc1*	mouse	NM_008625.2	102	TTGCACTTTGAGGGAAGCGA	CCTTGCCTGATGCCAGGTTA

*Hprt*: hypoxanthine-guanine phosphoribosyltransferase, *Cd*: cluster of differentiation, *Cpt*: carnitine palmitoyltransferase, *Cs*: citrate synthase, *Gclc*: glutamate-cysteine ligase catalytic subunit, *Gpx*: glutathione peroxidase, *Gsr*: glutathione-disulfide reductase, *Gsto*: glutathine S-transferase omega, *Sod*: superoxide dismutase, *Nos2*: nitric oxide synthase 2, *Mrc1*: mannose receptor C-type 1.

## Data Availability

The datasets used and/or analysed during the current study are available from the corresponding author upon reasonable request.

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
