# Peer review of "Long-Term High-Fat Diet Limits the Protective Effect of Spontaneous Physical Activity on Mammary Carcinogenesis"

_ijms, 2024, doi:10.3390/ijms25116221_

Round 1

Reviewer 1 Report

Comments and Suggestions for Authors

An interesting work aimed at finding a connection between the characteristics of a diet (short-term and long-term high-fat diet), physical activity and breast cancer. There are comments and suggestions on this work.

1) The authors clearly indicate in the introduction of the work that obesity is a risk factor for breast cancer. However, this is not entirely true. Currently, the involvement of BMI in the predisposition to breast cancer has been proven in numerous scientific studies, but at the same time there is an obvious inconsistency the results obtained. On the one hand, large-scale epidemiological studies (including meta-analyses) have convincingly shown the importance of increased BMI for the development of breast cancer [PMID: 18280327, PMID: 25129328, PMID: 27551723, PMID: 29403312,PMID: 36289879]. On the other hand, there is equally convincing evidence of a negative BMI relationship with breast cancer [PMID: 27551723PMID: 31243447, ]. A hypothesis has been put forward about the role of a higher BMI in the development of breast cancer in postmenopausal women, whereas premenopausal women, on the contrary, have an increased BMI is a protection factor against breast cancer [PMID: 29403312,PMID: 31243447]. It is believed that postmenopausal women with high BMI has large fat reserves, which cause a high concentration of estrogens in the body, which leads to a risk factor for developing breast cancer [PMID: 29403312,PMID: 31243447]. However, a high BMI in premenopausal women determines a longer anovulatory cycle, which leads to low levels of progesterone and estrogen, which leads to a protective effect in breast cancer in these women [PMID: 29403312,PMID: 31243447].Of course, these features of the obesity (BMI) relationship should be reflected both in the introduction of the work and in the discussion of the results obtained.

2) the authors indicate in the introduction of the work the involvement of genetic factors in the development of breast cancer. With this in mind, in the discussion of the work, it is necessary to consider in detail the possible relationships between the features of diet \ lipid metabolism (BMI) / physical activity and genetic factors in the formation of breast cancer. Also in the discussion, attention should be paid to the possible modifying role of diet\lipid metabolism (BMI)/ physical activity on the manifestation of a genetic predisposition to breast cancer, including genes associated with a hormonal profile (for example, a protein transporting sex hormones (SHBG),https://doi.org/10.3390/biomedicines12040818) and others (matrix metalloproteinases,PMID: 36289879 ), etc.

3. Can there be a connection between diet and physical activity with breast cancer of different directions in different periods of a woman's life (taking into account the multidirectional effect of obesity on breast cancer in the pre- and postmenopausal periods)?

4. It is recommended to indicate the limitations of this study.

Reviewer 2 Report

Comments and Suggestions for Authors

The authors present the article entitled "Long-term high-fat diet limits the protective effect of spontaneous physical activity on mammary carcinogenesis" in which they try to establish a relationship between the risk of developing breast cancer and the preventive effect of exercise.
The article is well written and easy to follow and understand, just some English language corrections should be made.

However, I have some observations for the authors:
- The main consideration I have is to know why the authors did not use a control group that did not have a diet rich in fatty acids. The mice in both groups present similar physical activity, however, it is not tested whether the physical activity in mice without a high-fat diet has the same physical activity, which would invalidate the researchers' conclusions in the sense that the high-fat diet Fat limits the protective effect of exercise on the development of breast cancer. Also with this control group we could see what amount of exercise is normal and what are the normal biochemical parameters.

- The authors mention that the results of biochemical tests are not associated with the development of metabolic syndrome, probably due to the time of exposure to the diet provided. However, these biochemical tests could be very different if they were compared to a group of mice on a normal diet.

- The authors mention "Before tumor cell implantation, the duration of diet had no impact on the daily distance traveled (LT HFD = 1.065 ± 0.076 vs ST HFD = 0.645 ± 0.082 km/day/mouse). Similarly, the tumor growth did not affect the distance traveled whatever the group (LT HFD= 0.803±0.045 vs ST HFD = 0.965 ± 0.217 km/day/mouse). "According to the aforementioned data, at first glance, it would seem that the duration of the diet does impact the The distance traveled between the 2 groups LT= 1.065 vs ST=0.645, is a difference of almost 40%, my doubt is because the authors mention that there is no difference.

- The authors mention "Among antioxidant enzymes, the total tissue activity of glutathione peroxidase (GPx) was not altered by the diet duration (LT HFD = 30.82 ± 20.72 vs St HFD = 3.46 ± 1.35 216
UI, Figure 6.A)." I consider that if there is a big difference 30.82 vs 3.46, despite the standard deviation, are these data correct?

- In the discussion the authors mention "This animal model of aged ovariectomized mice, fed with a hypercaloric diet, mimicking menopause and the risk of developing breast cancer, has been previously characterized and validated [23,28,29]. The EO771 tumor cells represent a luminal B cancer type [30,31] and are particularly immunogenic [23]. In this model, the high-fat diet was associated with a change in body composition and, in particular, fat gain without a change in lean mass compared. to a normocaloric diet and was of course associated with higher tumor progression (laboratory data). "In this sense, it would be convenient for the authors to present these data as supplementary material. Furthermore, they mention that the cell line is particularly immunogenic, so are the results observed in terms of the immune response as a result of the transplantation of the cell line, by itself, or are they a consequence of the duration of the diet?

Those are all my observations, again, my main concern is not having had a control group with a normocaloric diet.

Comments on the Quality of English Language

Minor editing of English language required

Round 2

Reviewer 1 Report

Comments and Suggestions for Authors

The adjustments made by the authors do not fully take into account the comments made:

1. the contribution of genetic factors in the development of breast cancer is not 5-10%, as the authors indicate, but much more: Firstly, according to large-scale twin studies performed in European populations and including materials on several tens [Möller, S.;  et al. The Heritability of Breast Cancer among Women in the Nordic Twin Study of Cancer. Cancer Epidemiol. Biomarkers Prev. 2016, 25, 145-150. doi: 10.1158/1055-9965.EPI-15-0913.] and hundreds [Mucci, L.A.; et al. Nordic Twin Study of Cancer (NorTwinCan) Collaboration. Familial Risk and Heritability of Cancer Among Twins in Nordic Countries [published correction appears in JAMA. 2016 Feb 23;315(8):822]. JAMA 2016, 315, 68–76. https://doi.org/10.1001/jama.2015.17703.] of thousands of twins pairs, the contribution of "genetics" to the BC development is 31%. Secondly, over 25% of hereditary cases of the disease are caused by mutations in highly penetrant (BRCA1,BRCA2,PTEN,TP53,CDH1,STK11) (increase the risk of developing BC up to 80%) and 2-3% in moderately penetrant (CHEK2,BRIP1,ATM,PALB2) (cause 2-fold in-creased BC risk) [Shiovitz, S; Korde, L.A. Genetics of breast cancer: a topic in evolution. Ann Oncol. 2015; 26, 1291-1299. doi:10.1093/annonc/mdv022.]. Thirdly, the results of large-scale GWAS showed associations with the disease of over 220 polymorphic loci of numerous candidate genes [GWAS catalog] and these GWAS SNPs "explain" 18% of the heritability of BC [Michailidou, K.; et al. Association analysis identifies 65 new breast cancer risk loci. Nature 2017, 551, 92–94. https://doi.org/10.1038/nature24284.]. Only 5% cases of the disease are associated with mutations in highly and moderately penetrant genes [Lilyquist, J.; et al. Common Genetic Variation and Breast Cancer Risk-Past, Present, and Future. Cancer Epidemiol. Biomark. Prev. 2018, 27, 380–394. https://doi.org/10.1158/1055-9965.EPI-17-1144.].

2. The authors indicate that "the majority are induced by sub-optimal lifestyle habits and environmental factors, as revealed by numerous clinical and epidemiological studies [3]. ", but at the same time cite one source in the link, which indicates the role of BMI alone in the development of breast cancer (Chen, Y.; et al Body Mass Index Had Different Effects on Premenopausal and Postmenopausal Breast Cancer Risks: A Dose-Response Meta-Analysis with 3,318,796 Subjects from 31 Cohort Studies. BMC Public Health 2017, 17, 936, doi:10.1186/s12889-017-4953-9.), which does not correspond to the statement made by the authors.

3. The authors indicate that "Only a small proportion of breast cancer, 5 to 10%, is hereditary and involves notably the mutation of the brca genes [2].". This statement applies only to the role of high- and medium- penetrant mutations in the formation of breast cancer, but in no way describes the role of all hereditary factors in the development of the disease (the data are given in comment No. 1).

The authors should make adjustments to the article taking into account the above comments and not present erroneous data in the work (including on the contribution of genetic factors), which significantly reduces the quality of the presented work. The authors should not "adjust" the literary data to their ideas, but on the basis of a full-fledged diverse analysis of the literary data, a more in-depth analysis of the interesting data obtained in the work should be carried out.

Reviewer 2 Report

Comments and Suggestions for Authors

The authors responded to my comments and observations, therefore, I consider that the article can be considered for publication.

Author Response

We would like to thank the reviewer for accepting this study following our responses.

Round 3

Reviewer 1 Report

Comments and Suggestions for Authors

the work can be accepted for publication in this edition.